# Genome-wide characterization of *WRKY* gene family in *Helianthus annuus* L. and their expression profiles under biotic and abiotic stresses

**Juanjuan Li**[1], **Faisal Islam**[2], **Qian Huang**[2], **Jian Wang**[2], **Weijun Zhou**[2], **Ling Xu**[1]*, **Chong Yang**[3]*

**1** Zhejiang Key Lab of Plant Secondary Metabolism and Regulation, College of Life Sciences and Medicine, Zhejiang Sci-Tech University, Hangzhou, China, **2** Institute of Crop Science, Ministry of Agriculture and Rural Affairs Lab of Spectroscopy Sensing, Zhejiang University, Hangzhou, China, **3** Bioengineering Research Laboratory, Institute of Bioengineering, Guangdong Academy of Sciences, Guangzhou, China

\* lxu@zstu.edu.cn (LX); parker815@163.com (CY)

**Data Availability Statement:** All relevant data are within the paper and its Supporting Information files.

## Abstract

WRKY transcription factors play important roles in various physiological processes and stress responses in flowering plants. Sunflower (*Helianthus annuus* L.) is one of the important vegetable oil supplies in the world. However, the information about *WRKY* genes in sunflower is limited. In this study, ninety *HaWRKY* genes were identified and renamed according to their locations on chromosomes. Further phylogenetic analyses classified them into four main groups including a species-specific WKKY group. Besides, *HaWRKY* genes within the same group or subgroup generally showed similar exon-intron structures and motif compositions. The gene duplication analysis showed that five pairs of *HaWRKY* genes (*HaWRKY8/9*, *HaWRKY53/54*, *HaWRKY65/66*, *HaWRKY66/67* and *HaWRKY71/72*) are tandem duplicated and four *HaWRKY* gene pairs (*HaWRKY15/82*, *HaWRKY25/65*, *HaWRKY28/55* and *HaWRKY50/53*) are also identified as segmental duplication events, indicating that these duplication genes were contribute to the diversity and expansion of *HaWRKY* gene families. The dN/dS ratio of these duplicated gene pairs were also calculated to understand the evolutionary constraints. In addition, synteny analyses of sunflower *WRKY* genes provided deep insight to the evolution of *HaWRKY* genes. Transcriptomic and qRT-PCR analyses of *HaWRKY* genes displayed distinct expression patterns in different plant tissues, as well as under various abiotic and biotic stresses, which provide a foundation for further functional analyses of these genes. Those functional genes related to stress tolerance and quality improvement could be applied in marker assisted breeding of the crop.

## Introduction

The *WRKY* gene family is considered as one of the largest transcription factor (TF) family in higher plants [1], which basically contain an approximate 60-residue DNA-binding domain, named as WRKY domain, with a highly conserved heptapeptide motif WRKYGQK and a

**Funding:** This work was supported by the National Natural Science Foundation of China (31701333), GDAS' Project of Science and Technology Development (2020GDASYL-20200103062), Zhejiang Provincial Natural Science Foundation (LGN18C130007), the Jiangsu Collaborative Innovation Center for Modern Crop Production, and Inner Mongolia Science & Technology Plan (201802072). The funders had no role in the design of the study and collection, analysis, and interpretation of data and in writing the manuscript.

**Competing interests:** The authors have declared that no competing interests exist.

$C_2H_2$- or $C_2HC$-type of zinc-finger motif included. Both the heptapeptide motif and zinc-finger motif are needed for binding of WRKY TFs to the *cis*-acting element W-box (C/T)TGAC (C/T) [2, 3]. *WRKY* gene family can be classified into three main groups (I-III), based on the number of WRKY domains and the structure of their zinc-finger motifs [4]. The group I WRKY proteins consist of two WRKY domains, whereas groups II and III contain only one. The group II and III WRKY proteins are distinguished by the type of zinc-finger motif, with a $C-X_{4-5}-C-X_{22-23}-H-X_1-H$ type of motif in group II and a $C-X_7-C-X_{23}-H-X_1-C$ type in group III [5].

The first *WRKY* gene was cloned and identified from sweet potato, encoding a 549 amino acid protein called SPF1 (SWEET POTATO FACTOR1) [6]. Since then, a large number of *WRKY* genes have been discovered from different plants. Functional analyses showed that *WRKY* genes are associated with various aspects of physiological processes, including seed dormancy and germination, root development, leaf senescence, modulation of flowering time, plant nutrient utilization etc [7]. The knockout mutant of *AtWRKY2* resulted in hypersensitivity of *Arabidopsis* to ABA during seed germination and post-germination, early growth, suggesting that *AtWRKY2* mediates seed germination and post-germination development [8]. Overexpression of *OsWRKY31* in rice inhibited plant lateral root formation and elongation, and also affected the transport process of auxin [9]. *AtWRKY12*, *AtWRKY13* and *AtWRKY71* are three main genes regulating *Arabidopsis* flowering time, with *AtWRKY12* and *AtWRKY13* working antagonistically under short daylight conditions [10], and *AtWRKY71* accelerating flowering [11]. It has been also documented that 12 *WRKY* genes are involved in leaf senescence in *Arabidopsis* and rice, as their mutants inhibited or promoted leaf senescence to different extents [7].

In addition to plant growth and development, *WRKY* genes also participate in modulation of plant tolerance to abiotic and biotic stress. Qiu and Yu [12] reported that overexpression of *OsWRKY45* in *Arabidopsis* significantly increased the expression level of *PR* genes and ABA/ stress regulated genes, thus contributed to the enhancement of disease resistance and salt and drought tolerance of the plant. *GmWRKY54* from soybean, which was confirmed in a DNA binding assay that could interact with the W-box, conferred salt and drought tolerance to transgenic *Arabidopsis*, possibly through the regulation of *DREB2A* and *STZ/Zat10* [13]. In tobacco, overexpression of grape *VvWRKY2* reduced the susceptibility to fungal pathogens like *Botrytis cinerea*, *Pythium* spp. and *Alternaria tenuis* [14]. The *WRKY1* in tobacco could be phosphorylated by a salicylic acid-induced protein kinase (SIPK), resulting in enhanced DNA-binding activity to a W-box sequence from the tobacco chitinase gene CHN50, and subsequently formation of hypersensitive response-like cell death [15]. Thus, *WRKY* genes may be involved in mitigating the damage caused by stresses, through interacting with the *cis*-element W-box and activating downstream plant defense signaling [2].

Common sunflower (*Helianthus annuus* L.) is grown throughout the world as an industrial crop for edible oil. It is the fourth important oilseed crop which contributes to 12% of the edible oil produced globally. However, sunflower production has been threatened by different stresses, among which drought and salinity are two major abiotic constraints [16]. Moreover, parasitic weed *Orobanche cumana* is a new emerged biotic issue worldwide [17]. WRKY transcription factors are involved in regulation of plant tolerance to both abiotic and biotic stresses. Thus, it is of great interest to characterize a *WRKY* gene family in sunflower and identify their functions under different stresses.

The *WRKY* gene family has been well studied in sunflower. Giacomelli et al. [18] have identified a total number of 97 *WRKY* genes in the Asteraceae family, while only 26 of them belong to *H. annuus*, and this identification was all based on EST database. The publication of reference genome will provide an opportunity to reveal the organization, expression and

evolutionary traits of common sunflower *WRKY* gene family at the genome-wide level. Badouin et al. [19] reported a high-quality reference for the sunflower genome (3.6 gigabases), with 17 chromosomes and 52,232 protein-coding genes on them. Guo et al. [20] identify 112 sunflower *WRKY* genes from this reference genome and Liu et al. [21] have extended this family to 119 members. In the current study, another sunflower database from a different sunflower genotype was used to search *WRKY* genes as support and addition to the previous works. A total of 90 *HaWRKY* genes were identified, among which 89 had corresponding genes in the updated sunflower *WRKY* gene family [21], whereas the rest one was newfound (S3 Table). The 90 *WRKY* genes could be classified into four main groups, including an extra WKKY group. Analyses on exon-intron organization, motif composition, gene duplication, chromosome distribution, phylogenetic relationship and gene synteny were further conducted to systemically characterize these common sunflower *WRKY* genes. Additionally, the expression patterns of *HaWRKY* genes in different plant tissues and in responses to different abiotic and biotic stresses were also recorded, to identify the implication of specific *WRKY* genes in different biological processes. The present findings provide a foundation for future research on functional characterization of *WRKY* genes in common sunflower.

## Materials and methods

### Gene identification

The genome of *H. annuus* (HA412.v1.1.bronze) was downloaded from Sunflower Genome Database (https://www.sunflowergenome.org/). The protein sequences of the WRKY family of *A. thaliana* were obtained from Plant Transcription Factor Database (http://planttfdb.cbi.pku.edu.cn/index.php) [22], which were used to search the *WRKY* genes from *H. annuus* genome via BlastP and tBlastN (E-value $\leq$ 1e-20). Then Pfam database (http://pfam.xfam.org/) and SMART database (http://smart.embl-heidelberg.de/) were used for verification of the WRKY domains [23, 24]. These potential sequences were further queried in the NCBI Conserved Domains Database (https://www.ncbi.nlm.nih.gov/Structure/cdd/cdd.shtml) and InterProScan Database (http://www.ebi.ac.uk/interpro/search/sequence-search) to validate the conserved domain [25, 26]. The molecular weight (Mw) and isoelectric point (pI) of the full-length proteins were predicted using the pI/Mw tool (https://web.expasy.org/compute_pi/) in ExPASy [27].

### Phylogenetic analysis and gene structure

Multiple sequence alignment based on WRKY domain sequences were conducted by clustal W analysis with default parameters. A neighbor-joining (NJ) tree was constructed in MEGA 5.2 with the following criteria: Poisson model, pairwise deletion, and 1000 bootstrap replications. Further maximum likelihood (ML) analysis of *WRKY* gene family from sunflower and Arabidopsis was conducted, to confirm the reliability of the result. The intron-exon structures of sunflower *WRKY* genes were analyzed by comparing predicted coding sequences with their corresponding full-length sequences using the online tool Gene Structure Display Sever (GSDS, http://gsds.cbi.pku.edu.cn/) [28]. The MEME online program (Multiple Expectation Maximization for Motif Elicitation) version 4.11.1 (http://meme-suite.org/index.html) was used to identify conserved motifs in the sunflower WRKY proteins [29].

### Chromosomal distribution and gene duplication

Multiple Collinearity Scan toolkit (MCScanX) was adopted to analyze the gene duplication events with default parameters [30]. PAL2NAL v14 was subsequently used to calculate dN and dS, with a dN/dS ratio of 1 indicative of neutral selection [31].

## Plant materials, growth conditions and treatments

Seeds of both *O. cumana* and two sunflower cultivars JY207 and TK0409 were provided by the Institute of Plant Protection, Inner Mongolia Academy of Agricultural and Animal Husbandry Sciences, Hohhot, China. For abiotic stresses, common sunflower cultivar TK0409 was used. The seeds were germinated and grown in peat moss according to our previous study [32]. At the four-leaf stage, seedlings with uniform size were selected to expose to NaCl (0, 150, and 300 mM) for salt stress and polyethylene glycol-6000 (PEG-6000, 0, 10% and 20% w/v) for simulated drought stress, respectively. The treatment concentrations were selected based on our preliminary experiments [32]. All the plants were placed in a growth chamber with light intensity ranging from 250 to 350 μmol m$^{-2}$ s$^{-1}$, temperature at 16–20°C, and relative humidity at approximately 55–60%. After another one week, roots and leaves of the sunflower seedlings were sampled for RNA isolation. For biotic stress, root parasitic weed *Orobanche cumana* were applied to common sunflower cultivars TK0409 (susceptible) and JY207 (resistant). 200 mg of *O. cumana* seeds were homogeneously mixed with 0.5 kg of the peat and vermiculite (1:1, v/v) substrate. Sunflowers were grown in the substrate containing *O. cumana* seeds as mentioned above. All the plants were placed in a growth chamber with 20°C at the daytime and 14°C at night, photoperiod for 14 h, and an irradiance of 300 μmol m$^{-2}$ s$^{-1}$. Three weeks after inoculation, sunflower roots were collected for experiments. In order to avoid tissue contamination of *O. cumana*, sunflower roots that were over 1 cm adjacent to the interaction site with *O. cumana* were harvested. Each treatment was replicated three times.

## Gene expression analysis

To determine the expression profiles of the *HaWRKY* genes under natural conditions, the transcriptomic data from 10 sunflower tissues, including bract, corolla, leaf, ligule, ovary, pollen, seed, stamen, stem, and style, were downloaded from the "Gene Expression Browser" of Sunflower Genome Database. Data were transformed by log$_2$ (FPKM+1). For biotic stress, the transcriptomic data were from our previous work [33]. Genes with false discovery rate (FDR) less than 0.01 and fold change more than 2 were considered differentially expressed between infected and non-infected sunflower. The expression profile heat-maps were generated using HemI 1.0 software.

For abiotic stress, TaKaRa MiniBEST Plant RNA Extraction Kit (Takara Bio, Kyoto, Japan) was used to extract total RNA from sunflower leaves and roots. The quantity and quality of RNA samples were assessed by agarose gel electrophoresis and using a Nanodrop 2000 Spectrophotometer for A260/A280 ratio. 200 ng of total RNA was reverse transcribed by using TaKaRa PrimeScript$^{TM}$ RT reagent Kit with gDNA Eraser. The gene-specific primers of sunflower for qRT-PCR amplification were designed by using Primer Premier 5.0 software and provided in S1 Table. SYBR Premix Ex Taq II (Tli RNaseH Plus, TaKaRa) in CFX96TM Real-Time PCR detection System (Bio-Rad, Hercules, CA, USA) was used to conduct the qRT-PCR experiments. The PCR conditions consisted of pre-denaturation at 95°C for 30 s, 40 cycles of denaturation at 95°C for 5 s, and annealing and extension at 58°C for 30 s. The default setting was used for the melting curve stage. The 2$^{-\Delta\Delta Ct}$ method with three replications was performed for analysis and the *ACT2* was selected as the reference gene. Genes with relative expression fold change (stress/control) $\geq$ 2 and $\leq$ 0.5 were considered significantly (Tukey's HSD test, $P < 0.05$) up and down-regulated, respectively.

## Ethics approval and consent to participate

*Helianthus annuus* cv. TK0409 and JY207 are widely cultivated and *Orobanche cumana* is a common parasitic weed in Inner Mongolia, China. Both *Helianthus annuus* and *Orobanche*

*cumana* are not listed in the appendices I, II and III of the Convention on the Trade in Endangered Species of Wild Fauna and Flora. Seeds of both *O. cumana* and two sunflower cultivars JY207 and TK0409 were collected in Bayannaoer, Inner Mongolia and provided by the Institute of Plant Protection, Inner Mongolia Academy of Agricultural and Animal Husbandry Sciences, Hohhot, China. Collection of plant materials complied with the institutional, national and international guidelines. No specific permits were required.

**Ethical approval.** This article does not contain any studies with human participants or animals performed by the authors.

## Results

### Identification of the *WRKY* genes

A total of 104 candidate *WRKY* genes were predicted from *H. annuus*, among which 14 were removed after domain check and the rest 90 were named as *HaWRKY1-HaWRKY90* (S2 Table). In comparison to the results of Guo et al. [20] and Liu et al. [21], one gene (*HaWRKY51*) in our study was found not included in the previous works, whereas the rest had their corresponding genes. The length of these genes ranged from 388 bp (*HaWRKY5*) to 8445 bp (*HaWRKY49*), with molecular weight (MW) from 10.48 to 74.25 kDa. The isoelectric point (pI) of these proteins ranged from 4.81 (HaWRKY86) to 10.44 (HaWRKY53) (S2 Table).

### Phylogenetic analysis of WRKY family members

Phylogenetic analysis of WRKY family members of *H. annuus* and *A. thaliana* was conducted based on WRKY domain. The common sunflower WRKY domains were divided into four large groups (Figs 1 and 2), corresponding to group I, II and III in Arabidopsis [2] and an extra WKKY group. In contrast, Guo et al. [20] and Liu et al. [21] just distributed WRKY genes into group I, II and III. Among 90 HaWRKY family members, group II accounts for the largest part with 48 HaWRKY proteins, followed by group I with 18 proteins and group III with 17 proteins (Fig 1). There were 7 HaWRKY proteins in group WKKY, with $C-X_5-C-X_{23}-H-X-H$ type zinc-finger motifs (Fig 2), which is not found in Arabidopsis (Fig 1). In addition, each group could be divided into several subgroups. Proteins with two WRKY domains were assigned as the N-terminal and the C-terminal WRKY domains according to their locations on protein. The proteins grouped either in N-terminal or C-terminal WRKY domains, usually followed by $C_2H_2$-type zinc-finger motifs ($C-X_4-C-X_{22-23}-H-X-H$), were classified as group I, with 16 identified as N-terminal WRKYs (I N) and 15 C-terminal (I C), and among them 13 members contained two WRKY domains (Fig 2). Group II of HaWRKY family could be clustered into five subgroups, with 4 in IIa, 10 in IIb, 13 in IIc, 11 in IId and 10 in IIe (Fig 1, S2 Table). 17 members of HaWRKYs in group III contained the $C-X_7-C-X_{23}-H-X-C$ type zinc-finger motifs (Fig 2, S2 Table), and were classified as subgroup IIIa. There were no HaWRKY proteins found in subgroup IIIb, not as that in *Arabidopsis* (Fig 1).

### Gene structure and motif composition of WRKY family members

The distributions of exons and introns on *HaWRKY* genes were investigated via GSDS program, to gain further insight into the structure diversity of the WRKY family in sunflower. As shown in Fig 3B, almost half number (43) of the *HaWRKY* genes had three exons, followed by 17 with two exons, 17 with four exons, 12 with five exons and 1 with seven exons. Genes within same groups generally shared similar structures, such as group IIIa, in which all *HaWRKY* genes possessed three exons and two introns. Most of the WRKY domains spanned an exon-exon junction, whereas *HaWRKY* genes with two WRKY domains in the group I at least had

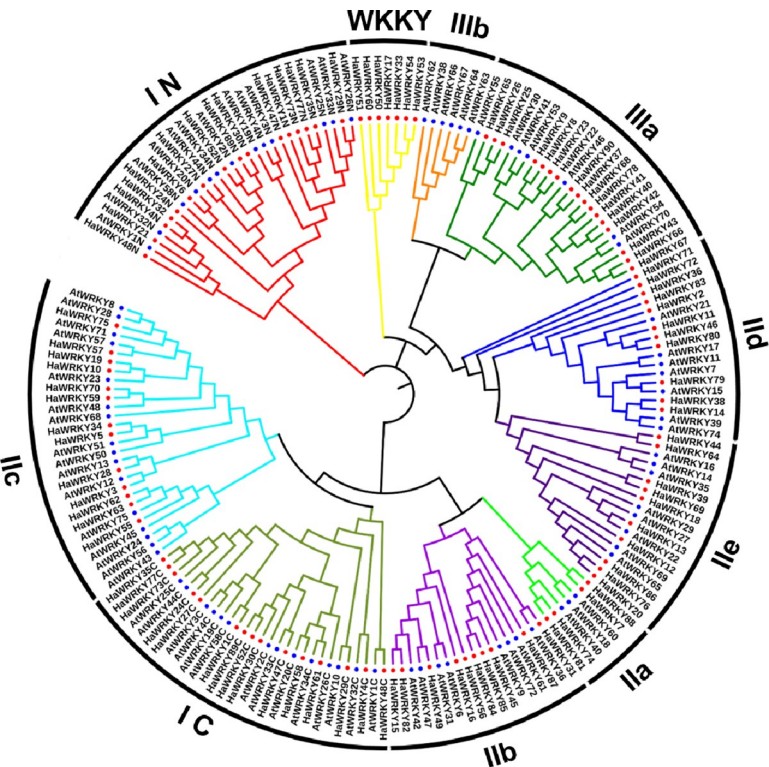

**Fig 1. Phylogenetic relationships of WRKY genes from common sunflower and Arabidopsis.** The different-colored braches indicate different groups (or subgroups). The red solid circles and blue solid squares represent WRKY genes from Arabidopsis and common sunflower, respectively.

one complete domain within one exon, except *HaWRKY1*. Further analyses on introns indicated that *HaWRKY* genes only with phase-0 introns (between two consecutive codons) were clustered into group IIa and IIb, and only with phase-2 introns (between the second and third nucleotide of a codon) into the group IId, IIe and IIIa. The phase-1 introns (between the first and second nucleotide of a codon) were widely distributed among these groups, except group IIa.

Motif structures on HaWRKY proteins were constructed via MEME program. As exhibited in Fig 3C, HaWRKY family members within same groups usually have similar motifs with similar arrangements. Motifs 1 and 6 are WRKY domains, with motif 6 only limited in group I, whereas motif 1 distributed all over the groups. Motifs 12, 15, 17 and 18 are unique to group IId, as well as motifs 11, 13 and 19 to group IIe, and motif 20 to group IIIa. In addition, the clustered HaWRKY pairs, like HaWRKY10/19, HaWRKY20/76, HaWRKY25/26, HaWRKY37/90, HaWRKY46/80, HaWRKY71/72, have similar protein lengths and same motif distributions, indicating the conserved motif structures of HaWRKY proteins within same groups.

## Evolution of group III *HaWRKY* genes

In order to understand the evolution of common sunflower group III *WRKY* genes, a phylogenetic tree of group III WRKY proteins from two monocots (rice and maize) and three dicots (sunflower, *Arabidopsis* and grape) was constructed. All the group III WRKY family members were divided into 10 clades as shown in Fig 4. WRKY proteins from closer species were clustered into same clades. Most proteins from dicots gathered in clade 1 and 3, whereas monocots

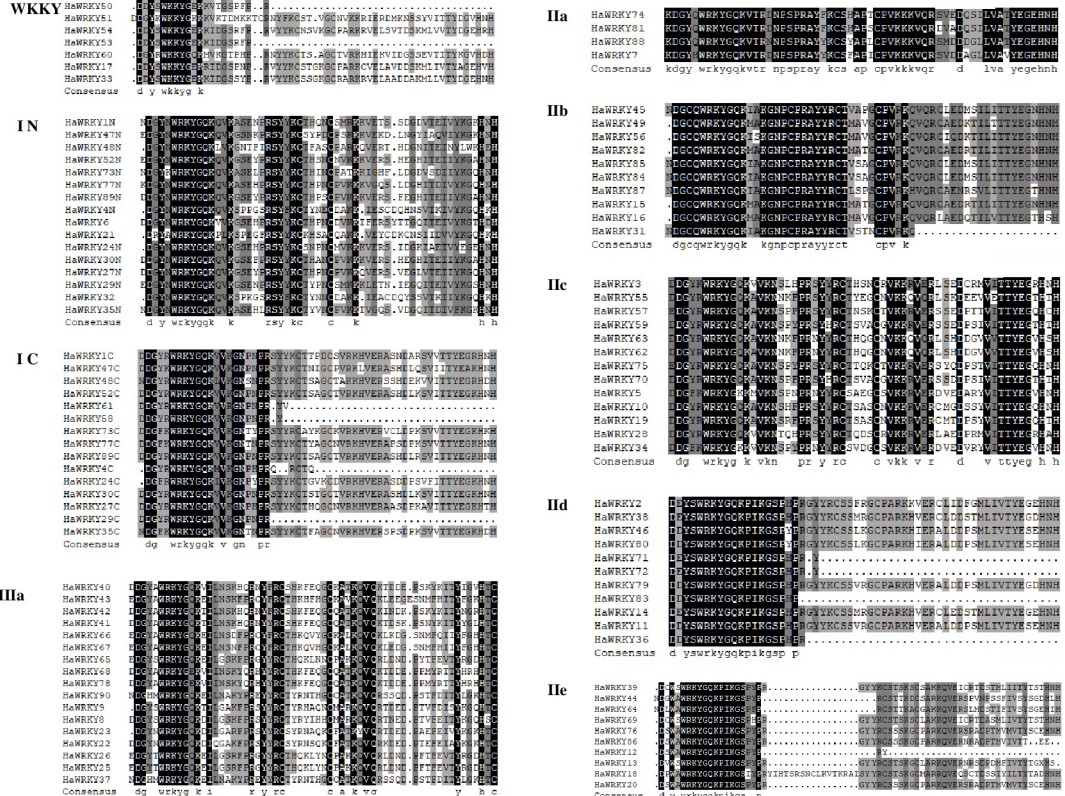

**Fig 2. Alignment of multiple HaWRKY protein sequences.** "N" and "C" indicate the N- and C-terminal of WRKY domains.

in clade 2, 4, 5, 6, 8, 9 and 10. Clade 7 contained proteins from all 5 species, indicating these proteins might be orthologues from a single ancestral gene.

MEME analysis was also conducted to search the conserved motifs of group III WRKY proteins from five species. Proteins within same clades usually displayed similar motif structures, indicating potential functional similarities among WRKY proteins. Motifs 1 and 7 were WRKY domains. Interestingly, motif 1 was found in all clades, whereas motif 7 was unique to clades 9 and 10, two clades only containing rice proteins, implying that motif 1 might have common function among different species, while motif 7 might play specific roles in rice and contribute to the divergence of group III *WRKY* genes. Motifs 1, 10 and 18 were specific to dicots. In contrast, motifs 12 and 19 were only observed in monocots. These motifs might be also important to the divergence of *WRKY* genes.

## Chromosomal location and synteny of *HaWRKY* genes

*HaWRKY* genes are distributed unevenly on 17 chromosomes (S2 Table, Fig 5A). Chromosomes Ha10 and Ha15 both have 13 *HaWRKY* genes as the largest groups, whereas there was no *HaWRKY* gene observed on chromosomes Ha2. No correlation between chromosome length and *HaWRKY* gene number could be determined.

Two or more genes located within 200 kb on same chromosome is defined as a tandem duplication event [34]. Five pairs of *HaWRKY* genes (*HaWRKY8/9*, *HaWRKY53/54*, *HaWRKY65/66*, *HaWRKY66/67* and *HaWRKY71/72*) are tandem duplicated on sunflower chromosomes Ha3, Ha12, Ha14 and Ha15. In addition, four segmental duplication events with four *HaWRKY* gene pairs (*HaWRKY15/82*, *HaWRKY25/65*, *HaWRKY28/55* and

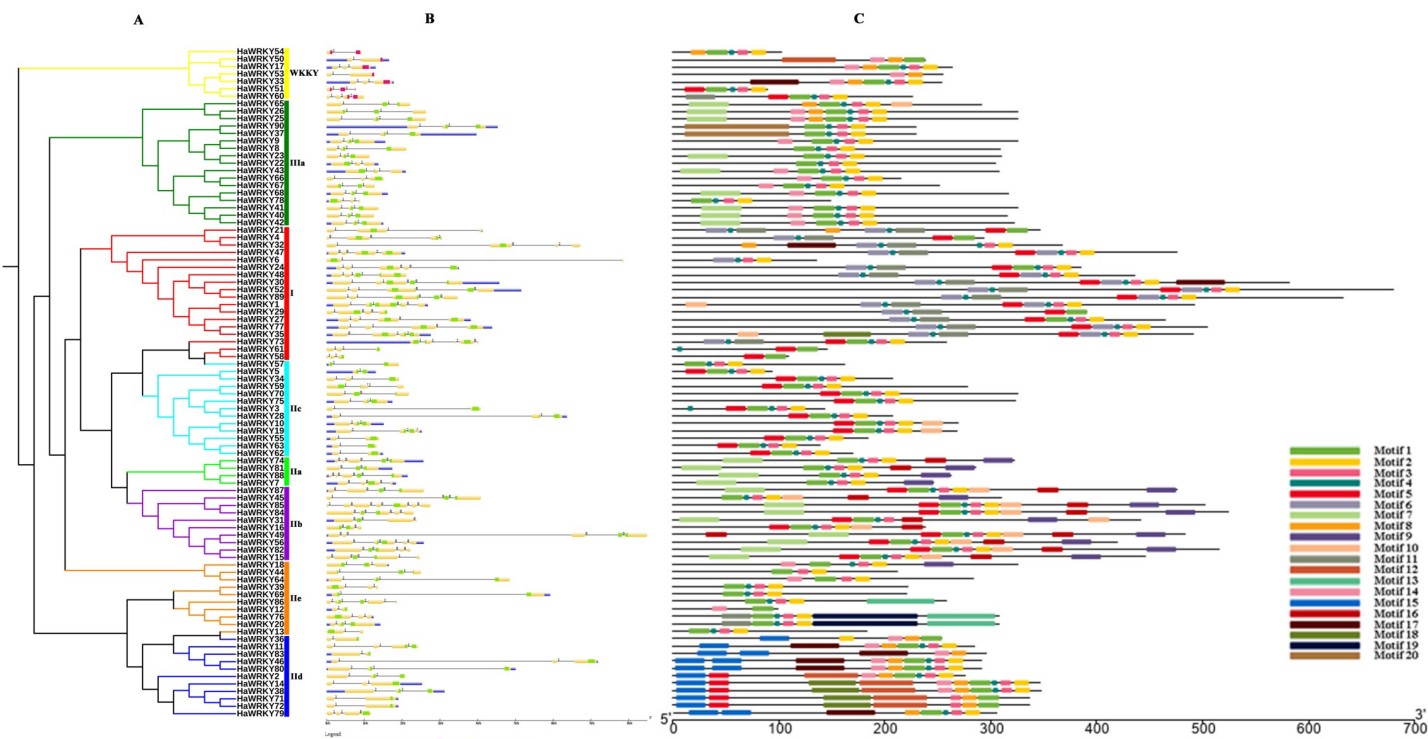

**Fig 3. Phylogenetic relationships, gene structures and motif compositions of WRKY genes in common sunflower.** (A) Phylogenetic tree of WRKY genes. The different-colored braches indicate different groups (or subgroups). (B) Exon-intron structures of WRKY genes. Blue boxes indicate 5' and 3' UTRs. Yellow boxes indicate exons. Black lines indicate introns. Red boxes indicate WRKY domains. The numbers indicate the phases of introns. (C) Motif compositions of WRKY proteins. Different motifs are displayed with different colored boxes.

*HaWRKY50/53*) are also identified (Fig 4). These results indicated that tandem and segmental duplication possibly contributes to the diversity and expansion of *HaWRKY* gene families. The dN/dS ratio of these duplicated gene pairs were calculated to understand the evolutionary constraints. The synonymous substitution rates (dS) of all segmental and tandem duplicated *HaWRKY* gene pairs were higher than non-synonymous substitution rate (dN) as shown in Table 1, indicating that *HaWRKY* gene family probably went through strong purifying selection during evolution.

Dual syntenies of common sunflower with *Arabidopsis* and rice were also conducted. A total of eight *HaWRKY* genes showed syntenic relationship with those in *Arabidopsis*, composing 9 orthologous pairs, whereas only one *HaWRKY* gene was collinear with one in rice (Fig 5B and 5C). Similarly, more collinear gene pairs were observed between sunflower and *Arabidopsis* than rice, as sunflower is phylogenetically closer to *Arabidopsis*. *HaWRKY16* was associated with two *Arabidopsis* genes and *HaWRKY25* and *HaWRKY65* are syntenic with a same *Arabidopsis* gene. *HaWRKY25* is also found to be syntenic with a rice gene, indicating that these orthologous pairs might occur before the divergence of monocots and dicots.

## Transcriptomic pattern of *HaWRKY* genes from different tissues

The transcriptome data of *HaWRKY* genes of different sunflower tissues were downloaded from Sunflower Genome Database. 20 of the 90 identified *HaWRKY* genes weren't expressed in all ten tissues (Fig 6A), which might be pseudogenes, have special temporal and spatial expression patterns or express in other tissues. The expression patterns of *HaWRKY* genes in sunflower were organ-specific, as bract, corolla, ligule, ovary, seed and stamen, which are

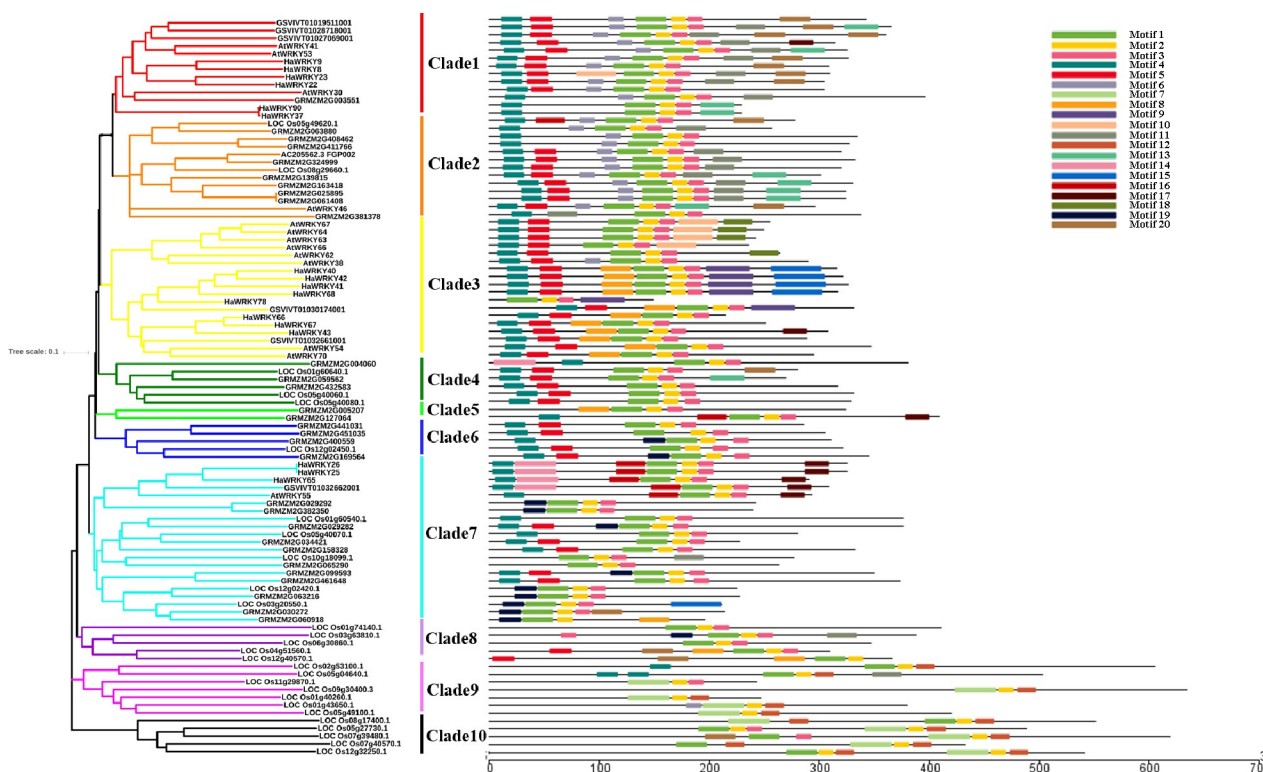

**Fig 4. Phylogenetic relationships and motif compositions of group III WRKY proteins from five monocot and dicot plants.** On the left side, proteins are clustered into 10 clades, marked with different colors. On the right side, different motifs are displayed with different colored boxes.

related to flower, were clustered into a big group, and leaves and stem were in another group (Fig 6A). Most of the *HaWRKY* genes didn't express in pollen (Fig 6A). In general, the expression levels of *HaWRKY* genes in bract, ligule, leaves and stem were higher than that in other tissues (Fig 6B). *HaWRKY17/22/79/81* displayed highest transcript abundances across all tissues except pollen and were clustered into a group, whereas the expression levels of *HaWRKY23/31/37/40/68/84* were extremely low in all tested tissues (Fig 6A). The expression patterns of some genes were tissue-specific, for example, *HaWRKY73* was abruptly induced only in leaves, *HaWRKY3* in stem, *HaWRKY11* in style, etc.

## Profiles of *HaWRKY* genes under abiotic and biotic stress

Twenty-three *HaWRKY* genes which were highly induced in different tissues of common sunflower (except in pollen) were selected to test the reactions of different *WRKY* genes to different abiotic stresses. Generally, *HaWRKY* genes were inhibited in sunflower leaves after treatment of PEG and NaCl with different concentrations, whereas *HaWRKY29/30* at 150 mM NaCl and *HaWRKY48/89* at 20% PEG were significantly ($P < 0.05$) up-regulated by 46%/140% and 70%/51% compared with control, respectively. But no any significant (Tukey, $P<0.05$) changes of *HaWRKY55/57* were observed in response to abiotic stresses in all comparisons (Fig 7A). Similarly, in roots as shown in Fig 7B, expression levels of most *HaWRKY* genes decreased significantly in response to simulated drought stress. For example, *HaWRKY13/14/16/57/77* were significantly ($P < 0.05$) down-regulated under 10% and 20% drought stress, while *HaWRKY29/74* were significantly ($P < 0.05$) down-regulated by 55% and 70% only at high concentration of PEG, as compared with control. The expression level of

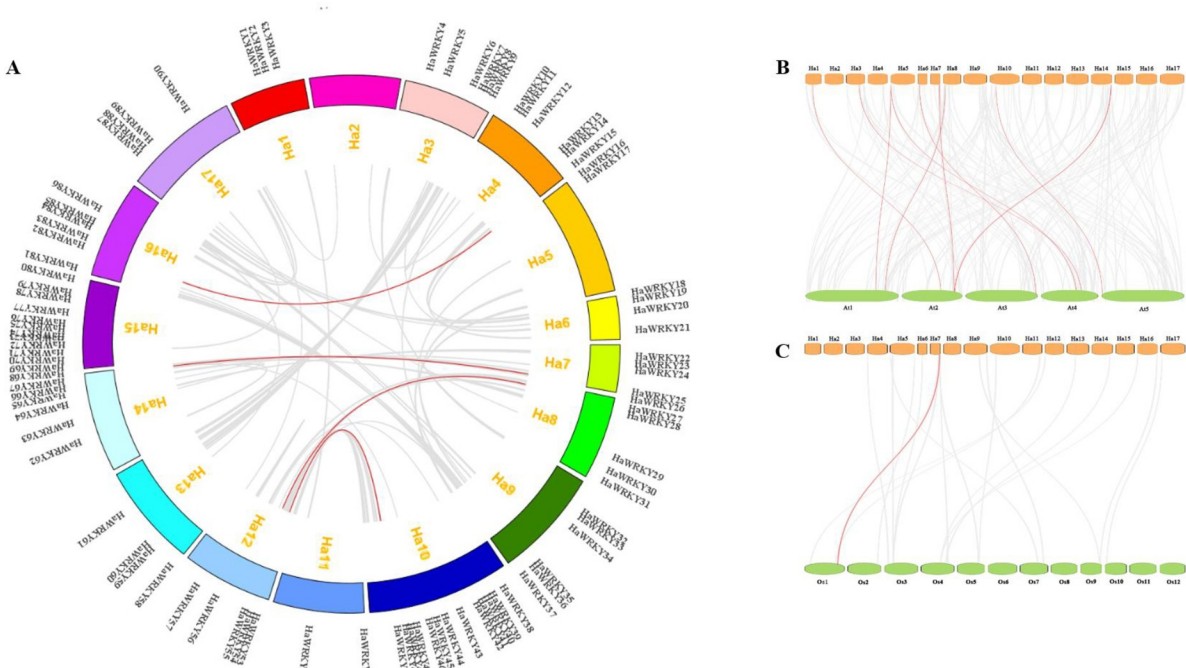

**Fig 5. Genome localization and synteny analyses of WRKY genes within common sunflower, and between common sunflower and two representative plant species.** (A) Chromosomal distribution and interchromosomal relationships of common sunflower WRKY genes. Gray lines indicate all sytenic gene pairs in common sunflower genome and red lines indicate duplicated WRKY gene pairs. (B-C) Synteny analyses of common sunflower WRKY genes with Arabidopsis and rice, respectively. Gray lines indicate collinear gene pairs between common sunflower and other plant genomes and red lines indicate syntenic WRKY gene pairs.

*HaWRKY52* and *HaWRKY89* were up-regulated by PEG, and 10% PEG significantly ($P < 0.05$) increased by 152% and 179% compared with control. In contrast, Most of the *HaWRKY* genes were significantly ($P < 0.05$) up-regulated after treatment with NaCl in roots, as compared to the control. Among them, the transcript levels of 14 *HaWRKY* genes, including *HaWRKY3/5/14/16/30/35/38/48/52/59/79/81/82* and *HaWRKY89* increased and 6 *HaWRKY* genes (*HaWRKY9/10/22/29/55/74*) decreased, as the concentration of NaCl went up. *HaWRKY13/57/77* were significantly ($P < 0.05$) depressed by treatment of NaCl in sunflower roots.

In order to understand the role of sunflower *WRKY* gene family against biotic stress, transcription levels of two contrasting common sunflower cultivars (TK0409, susceptible; JY207,

**Table 1. dN/dS analyses for the duplicated *WRKY* gene pairs of sunflower.**

| Duplicated gene 1 | Duplicated gene 2 | dN | dS | dN/dS | Purifying Selection | Duplicate type |
|---|---|---|---|---|---|---|
| *HaWRKY8* | *HaWRKY9* | 0.3047 | 0.8041 | 0.3790 | Yes | Tandem |
| *HaWRKY53* | *HaWRKY54* | 1.0653 | 15.1286 | 0.0704 | Yes | Tandem |
| *HaWRKY65* | *HaWRKY66* | 0.8347 | 55.0897 | 0.0152 | Yes | Tandem |
| *HaWRKY66* | *HaWRKY67* | 0.1132 | 0.265 | 0.4273 | Yes | Tandem |
| *HaWRKY71* | *HaWRKY72* | 1.0141 | 1.2005 | 0.8447 | Yes | Tandem |
| *HaWRKY15* | *HaWRKY82* | 0.1847 | 1.2239 | 0.1509 | Yes | Segmental |
| *HaWRKY25* | *HaWRKY65* | 0.2458 | 0.715 | 0.3437 | Yes | Segmental |
| *HaWRKY28* | *HaWRKY55* | 0.7764 | 59.8252 | 0.0130 | Yes | Segmental |
| *HaWRKY50* | *HaWRKY53* | 0.1218 | 0.6674 | 0.1825 | Yes | Segmental |

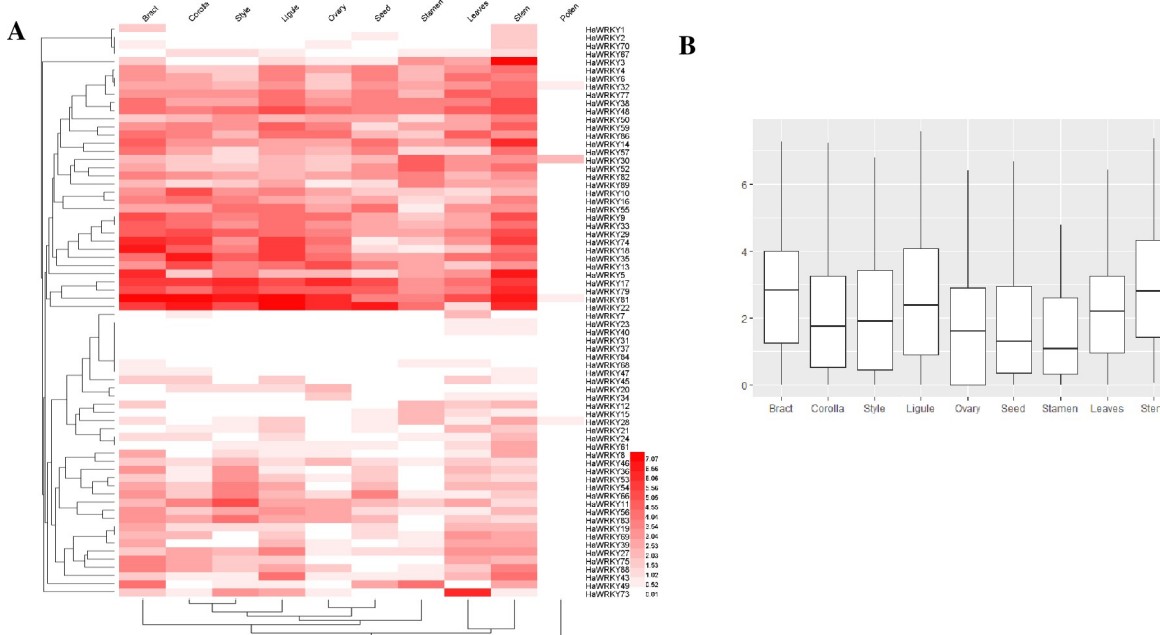

**Fig 6. Expression profile of WRKY genes in different tissues of common sunflower.** (A) Hierachical clustering of expression profile of WRKY genes from different tissues. Data were transformed with a log$_2$ (FPKM+1) transformation. (B) Boxplot of expression levels of WRKY genes in different tissues.

resistant) under infection of root parasitic weed *Orobanche cumama* were tested. Most of the genes were inhibited in both cultivars under attack of *O. cumana*, as compared to their corresponding cultivars without attack, respectively (Fig 8). Interestingly, *HaWRKY 7/15/44/45/68/71/72/76/85* were induced after attack of *O. cumana* in resistant cultivar JY207, whereas still depressed in susceptible cultivar TK0409 (Fig 8), suggesting these genes might partly contribute to the resistance of sunflower against *O. cumana*.

## Discussion

The WRKY transcription factor family is considered to be involved in diverse stress responses, developmental and physiological processes in plants. Systematical characterization of *WRKY* genes in several species has been studied, including *Arabidopsis*, rice, tomato, maize etc. Sunflower *WRKY* genes have been well characterized benefiting from the release of its reference genome in previous studies [20, 21]. However, there are two sunflower genome database available ("HanXRQr1.0" and "HA412.v1.1.bronze") assembled from different sunflower genotypes. Guo et al. [20] and Liu et al. [21] have used the "HanXRQr1.0" database for retrieval of *WRKY* genes. In our study, we used "HA412.v1.1.bronze" database to search *WRKY* genes as support and addition to the previous works. Indeed, we found 89 *WRKY* genes which have corresponding genes in the results of Guo et al. [20] and Liu et al. [21], and we discovered another *WRKY* gene (*HaWRKY51*) which was neglected in their works.

Multiple protein sequence alignments revealed domain variations in common sunflower WRKY family. In comparison to the results of Guo et al. [20] and Liu et al. [21] that classify sunflower WRKY proteins into three normal groups, an extra WKKY group with 7 proteins was identified in our study based on the phylogenetic analysis, including HaWRKY 60 with WKKYGQK, HaWRKY33/50/53/54 with WKKYGEK, and HaWRKY51 with WKKYGKK (Fig 2). Interestingly, this group has been also found in *Helianthus exilis*, *Helianthus petiolaris*

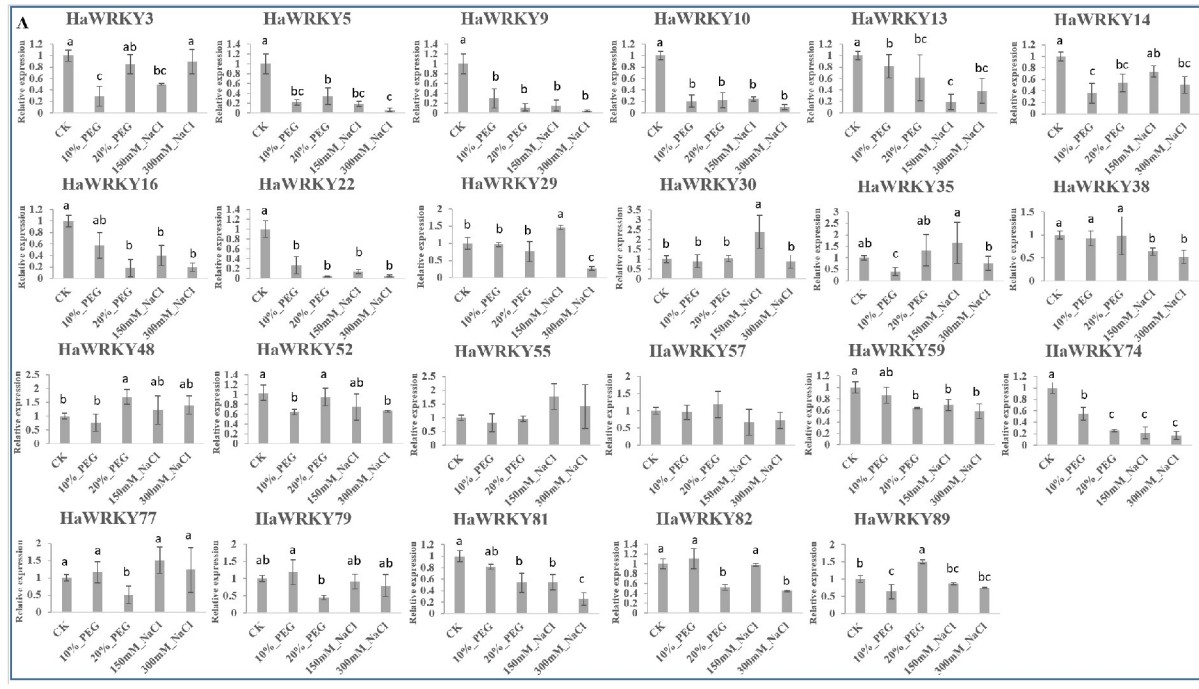

A

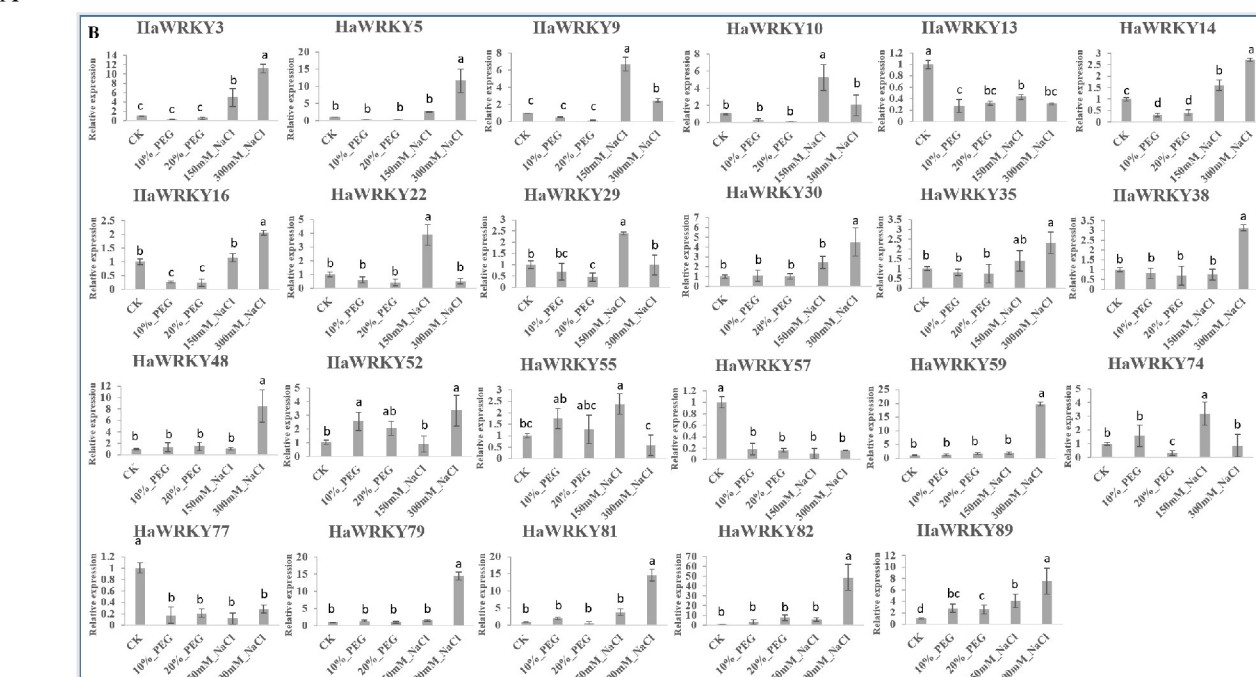

B

**Fig 7. Expression profile of 23 selected *HaWRKY* genes in responses to treatments of PEG and NaCl.** (A) Expression profile of WRKY genes in common sunflower leaves. (B) Expression profile of WRKY genes in common sunflower roots.

and *Helianthus tuberosus* [19], indicating that the WKKY variation is common in the Asteraceae. In addition, WKKYGQK, WKKYGKK and WKKYGEK are also observed in different legumes, but with low frequencies [35]. However, there are no more reports about WKKY group in other plant species. Although the WRKYGQK is highly conserved in most WRKY

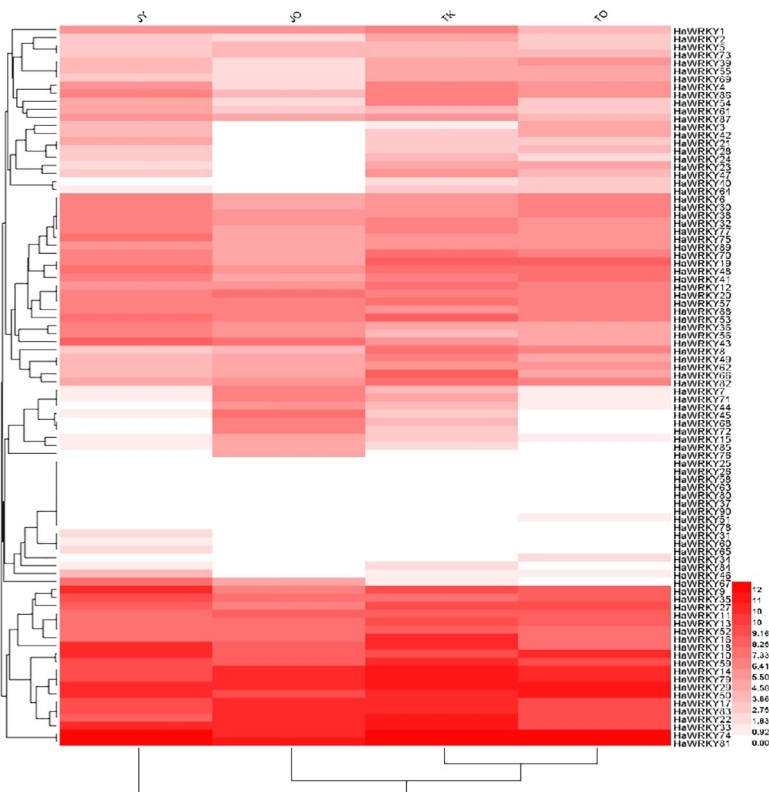

**Fig 8. Expression profile of *HaWRKY* genes in response to infection of parasitic weed *Orobanche cunama*.** JY represents resistant sunflower cultivar JY207; JO represents JY207 with infection of *O. cumana*; TK represents susceptible sunflower cultivar TK0409; TO representsTK0409 with infection of *O. cumana*.

domains, variation in the core sequence has been documented [35]. In our studies, mutations happened to R and Q sites, while the others were conserved. WRKYGKK and WRKYGEK are the most frequently occurring variants of the core sequence in most plant species [35]. As the WRKYGQK core sequence can interact with the W-box to activate downstream genes, the variations in this motif might influence the function of downstream target genes [13, 36]. Thus, further investigations on functions and binding specificities of these sunflower proteins with mutated WRKY motifs might provide deep insight into this transcription factor family.

A domain loss is common in the *WRKY* gene family in plants, which is recognized as a divergent force for expansion of this gene family [3, 37]. In the current study, 5 domain loss events were found in group I, suggesting a potential cause of the diversity of *WRKY* genes in this group. Tandem and segmental duplication events also played a pivotal role in the expansion of *WRKY* gene family [4]. Five pairs of tandem and four pairs of segmental duplicated genes were identified in the present study, with five pairs in group IIIa, two pairs in group IId, one pair in group IIb and one pair in group IIc. This result indicated that tandem and segmental duplication events might contribute to the amplification of sunflower *WRKY* genes in these groups, as compared to those of *Arabidopsis*.

The origin of *WRKY* genes from group III appears to have occurred prior to the divergence of monocots and dicots, and then numerous duplications and diversifications happened after that event [38]. In order to explore how the WRKY group III gene family evolved, a phylogenetic tree of WRKY group III proteins from sunflower with two dicots (*Arabidopsis*, grape) and two monocots (rice, maize) was constructed, which divided the 17 group III *HaWRKY*s

into three clades. WRKY proteins from closer species appeared to be clustered together. Both monocots and dicots proteins occurred in many clades, suggesting group III *WRKY* genes diversified before the monocot-eudicot split. In addition, clade 7 contained group III WRKY proteins from all 5 species, which tended to form monocot- and dicot-specific subclades, implying that group III *WRKY* genes evolved separately after the divergence of monocots and dicots.

It is well known that *WRKY* genes play essential roles in plant growth and development [7]. Li et al. [39] reported that *AtWRKY13* functioned in stem development, as a weaker stem phenotype was observed and lignin-synthesis-related genes were repressed in *Arabidopsis wrky13* mutants. In the current study, the orthologous of *AtWRKY13* in sunflower, *HaWRKY28* displayed high expression levels in stem, indicating that these genes might also act in stem development in sunflower. Overexpression of *WRKY15* exhibited an increased leaf area of *Arabidopsis*, which implied that *AtWRKY15* seemed to be involved in leaf growth [40]. *HaWRKY79*, which is the orthologous of *AtWRKY15*, were highly induced not just in leaves, but across all tissues, indicating that this gene might be constitutive in sunflower plant growth and development. In contrast, *HaWRKY7* was specifically expressed in leaves, suggesting its role in sunflower leaf growth. Among all *WRKY* genes in sunflower, only *HaWRKY30* displayed a high level of expression in pollen, with other *WRKY* genes extremely low expressed. Interestingly, two pollen-specific regulators in *Arabidopsis*, *AtWRKY34* and *AtWRKY2*, have phylogenetically close relationship with this sunflower *WRKY* gene, indicating that *HaWRKY30* might be associated with pollen developmental modulation.

In addition to their role in plant growth and development, WRKY TFs also play pivotal roles in various stress responses, providing an important basis for genetic improvement of crops. Drought and salinity, both of which can cause plant cellular dehydration [41], are two major constraints to sunflower production [42]. Responses of plants to drought and salinity usually result in accumulation of reactive oxygen species (ROS) and abscisic acid (ABA) [41, 43], which activate downstream *WRKY* genes [7].

Overexpression of a membrane-localized cysteine-rich receptor-like protein kinase, CRK5 in *Arabidopsis*, led to increase of ABA sensitivity and promotion of stomatal closure, and subsequent enhancement of plant drought tolerance. Knockout of *AtWRKY18*, *AtWRKY40* and *AtWRKY60* significantly increased the expression of *CRK5*, suggesting negative regulation of these three genes on *CRK5* [44]. In our study, the relative expression levels of two orthologous of *AtWRKY40*, *HaWRKY74* and *HaWRKY81*, were recorded in sunflower roots and leaves. Expression levels of both two genes decreased as the concentrations of PEG increased in sunflower leaves, while in sunflower roots, two genes were induced under low concentration of PEG and inhibited under high concentration. It has also been reported that *AtWRKY46*, *AtWRKY54*, and *AtWRKY70* are implicated in promotion of BR-regulated plant growth and inhibition of drought response, as reduced BR-regulated growth and higher survival rates under drought stress was observed in their triple mutant [45]. *HaWRKY9* and *HaWRKY22*, which were phylogenetically close to *AtWRKY46*, were both repressed under PEG treatments in sunflower roots and leaves. These results are suggesting that sunflower probably enhanced drought tolerance via down-regulating specific *WRKY* genes and subsequently activating downstream signal pathways. The increase of ABA level caused by drought usually induces high expression of *AtWRKY57*, which binds to W-box in the promoter region of the downstream response genes. *HaWRKY57*, the orthologous of *AtWRKY57*, displayed a high expression level under PEG treatment in sunflower roots. Interestingly, the increase of ROS level caused by salinity also activates *AtWRKY57*, and consistently, *HaWRKY57* was highly expressed under treatment of 300 mM NaCl in sunflower roots. These results indicated that *HaWRKY57* might share similar functions with *AtWRKY57* in sunflower under drought and

salinity. *AtWRKY15* is another *WRKY* gene induced by ROS, but will make *Arabidopsis* more susceptible to osmotic stress and oxidative stress [40]. In our study, *HaWRKY79*, the orthologous of *AtWRKY15*, was significantly suppressed in sunflower roots under treatment of NaCl, implying their similar roles in conferring salt tolerance.

The parasitic weed *Orobanche cumana* is a new emerged threat to sunflower production worldwide. Previous studies proposed that *O. cumana* deployed effectors in sunflower to suppress host defense responses and resistant sunflower cultivars recognized effectors with the help of R proteins to activate effector-triggered immunity. WRKY family has been found to be involved in the microbe-associated molecular pattern-triggered immunity, PAMP-triggered immunity or effector-triggered immunity [7]. Thus, it is worthy of studying *WRKY* genes in sunflower against *O. cumana*. According to our transcriptome data, *HaWRKY7/15/44/45/68/71/72/76/85* were specifically induced in sunflower resistant cultivar under attack of *O. cumana*, whereas repressed in susceptible cultivar during same interactions, indicating their potential roles in conferring resistance to sunflower against *O. cumana*.

## Conclusion

In this study, we identified 90 *WRKY* genes from *Helianthus annuus* L. and characterized their structure, duplication, chromosomal distribution, phylogenetic tree, followed by tissue-differential gene expression and differential expression in response to biotic and abiotic stress. *HaWRKY* genes within same group or subgroup generally showed similar exon-intron structures and motif compositions. Synteny analyses of sunflower *WRKY* genes provided deep insight to the evolution of *HaWRKY* genes. The expressions of *HaWRKY* genes suggested that most of the 23 selected *HaWRKY* genes were found to play an important role in regulating PEG and salt stress and 9 genes including *HaWRKY 7/15/44/45/68/71/72/76/85* from 90 *HaWRKY* genes have potential roles against *O. cumana* infection. Taken together, this study provides a good basis for further investigation of the biological functions and evolution of *HaWRKY* genes.

## Supporting information

**S1 Table. Primers for qRT-PCR.**
(DOCX)

**S2 Table. Basic information regarding the presence of WRKY genes in sunflower.**
(DOCX)

**S3 Table. Corresponding gene names.**
(DOCX)

## Acknowledgments

The authors thank to the Institute of Plant Protection, Inner Mongolia Academy of Agricultural and Animal Husbandry Sciences, Hohhot, China for providing plant materials.

## Author Contributions

**Conceptualization:** Ling Xu, Chong Yang.

**Formal analysis:** Juanjuan Li.

**Funding acquisition:** Weijun Zhou, Ling Xu.

**Investigation:** Juanjuan Li, Faisal Islam, Qian Huang, Jian Wang, Chong Yang.

**Project administration:** Weijun Zhou, Ling Xu.

**Software:** Juanjuan Li, Qian Huang, Jian Wang, Chong Yang.

**Validation:** Juanjuan Li, Faisal Islam.

**Visualization:** Juanjuan Li, Qian Huang, Jian Wang.

**Writing – original draft:** Juanjuan Li.

**Writing – review & editing:** Faisal Islam, Weijun Zhou, Ling Xu, Chong Yang.

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
