## [Decision Letter · Decision Letter 0]

9 Oct 2020

PONE-D-20-29683

Genome-wide characterization of WRKY gene family in Helianthus annuus L. and their expression profiles under biotic and abiotic stresses

PLOS ONE

Dear Dr. Ling Xu,

Thank you for submitting your manuscript to PLOS ONE. After careful consideration, we feel that it has merit but does not fully meet PLOS ONE’s publication criteria as it currently stands. Therefore, we invite you to submit a revised version of the manuscript that addresses the points raised during the review process.

Spellings and English language needs to be checked thoroughly. Overall, drafting of many sentences need to be improved. Tidying up the text is also suggested. I agree with the reviewer that some changes/clarification are needed in Conclusion and M&M section.

We look forward to receiving your revised manuscript.

Kind regards,

Basharat Ali, Ph.D

Academic Editor

PLOS ONE

Journal Requirements:

2. Please include captions for your Supporting Information files at the end of your manuscript, and update any in-text citations to match accordingly. Please see our Supporting Information guidelines for more information: http://journals.plos.org/plosone/s/supporting-information

Reviewers' comments:

Reviewer's Responses to Questions

**Comments to the Author**

1. Is the manuscript technically sound, and do the data support the conclusions?

Reviewer #1: Yes

Reviewer #2: Yes

2. Has the statistical analysis been performed appropriately and rigorously? 

Reviewer #1: Yes

Reviewer #2: Yes

3. Have the authors made all data underlying the findings in their manuscript fully available?

Reviewer #1: Yes

Reviewer #2: Yes

4. Is the manuscript presented in an intelligible fashion and written in standard English?

Reviewer #1: Yes

Reviewer #2: Yes

5. Review Comments to the Author

Reviewer #1: The authors conducted a genome-wide analysis of WRKY gene family in sunflower under drought, salt and weed stresses. The genes were characterized by phylogenetic analysis, gene structure and motif analysis, and expression analysis. Synteny analysis of the genes between sunflower and two other species, rice and Arabidopsis, were also done to understand the evolutionary aspects. The article is well-written. I would recommend publication upon minor revisions as indicated below:

Line 26: add a comma after “WKKY group”; add “the” before “same group”

Line 33: the last sentence could change into ”Those functional genes related to stress tolerance and quality improvement could be applied in marker assisted breeding of the crop”.

Line 86: add a comma after “abiotic constraints”

Line 92: Asteraceae should not be italic, please change into “Asteraceae family”

Line 167-169: the sentence “For biotic stress, common sunflower cultivars TK0409 (susceptible) and JY207 (resistant), and root parasitic weed Orobanche cumana were applied” is not clear. Do you mean: “For biotic stress, root parasitic weed Orobanche cumana were applied to common sunflower cultivars TK0409 (susceptible) and JY207 (resistant)”?

Line 194-203: please also indicate if house-keeping genes were used as reference for the qPCR, and if yes, what genes are used?

Line 332: please change “23” into “Twenty-three”

Line 370: please delete “fortunately”

Line 375: please delete “results” at the end of the line

Conclusion section: please also indicate here what HaWRKY genes (and/or how many) are the most important for each of the biotic or abiotic stresses.

Fig. 4 legend: “On the right side, Different motifs…” please change “Different” into “different”.

Reviewer #2: Comment of Reviewer

In the manuscript “Genome-wide characterization of WRKY gene family in Helianthus annuus L. and their expression profiles under biotic and abiotic stresses”, Li et al. identified WRKY family genes in sunflower and renamed them according to their locations on 25 chromosomes. They classified them into four main groups including a species-specific WKKY group using phylogenetic analyses, and they provided deep insight to the evolution of HaWRKY genes in sunflower by synteny analyses. Through transcriptomic and qRT-PCR analyses, they displayed distinct expression patterns of HaWRKY genes in different tissues, and under various abiotic and biotic stresses, which provide a foundation for further functional analyses of these genes.

This manuscript provides compelling evidence that HaWRKY family genes are associated with the development of the sunflower, as well as various abiotic and biotic stresses responsive, and provides a foundation for further functional analyses of HaWRKY genes. By using a non-standard genetic model like sunflower, the authors also help diversify studies in plant development. Please find my comments on the manuscript below.

1. Abstract is generally well-written but lacks some important data of the results.

2. Figure 7 and 8: Student’s t-test is fine here, but I suggest using Tukey’s HSD test to test all comparisons. This will help to show if 10% and 20% PEG, as well as 15mM and 30mM NaCl lines are different from each other, rather than just comparing to CK.

3. Line 53 “Functional analyses show that WRKY…” should be “Functional analyses showed that WRKY g…”.

4. In Materials and Methods part, some paragraph should could be more concise such as plant materials, growth conditions and treatments section.

5. The English writing should be substantially improved. Many sentences are illogical and cannot be understandable.

6. Overall: Add test statistics and p-values where significance is mentioned in text.

7. Overall: Italicize gene names.

6. PLOS authors have the option to publish the peer review history of their article (what does this mean?). If published, this will include your full peer review and any attached files.

Reviewer #1: No

Reviewer #2: No

---

## [Author Response · Author response to Decision Letter 0]

20 Oct 2020

Responses addressing Reviewers’ Comments

Comments of Reviewer #1: The authors conducted a genome-wide analysis of WRKY gene family in sunflower under drought, salt and weed stresses. The genes were characterized by phylogenetic analysis, gene structure and motif analysis, and expression analysis. Synteny analysis of the genes between sunflower and two other species, rice and Arabidopsis, were also done to understand the evolutionary aspects. The article is well-written. I would recommend publication upon minor revisions as indicated below:

Detailed comments:

Remark: Line 26: add a comma after “WKKY group”; add “the” before “same group”

Answer: Thanks for your kind comments. We have added a full stop after “WKKY group”; and added “the” before “same group” in Line 28. We also have tried our best to improve the Abstract Section in red front.

Remark: Line 33: the last sentence could change into “Those functional genes related to stress tolerance and quality improvement could be applied in marker assisted breeding of the crop”.

Answer: Many thanks for your kind suggestion. The last sentence has been changed into “Those functional genes related to stress tolerance and quality improvement could be applied in marker assisted breeding of the crop” expression in Line 41-42.

Remark: Line 86: add a comma after “abiotic constraints”

Answer: Thanks for your kind suggestion. We have added a comma after “abiotic constraints” in Line 93, and added “Moreover,” after the comma to improve the English.

Remark: Line 92: Asteraceae should not be italic, please change into “Asteraceae family”

Answer: Thank you for your kind comment. In the revised manuscript, “Asteraceae” has been changed into “Asteraceae family” in Line 99 in red font. Moreover, the italic “Asteraceae” was also changed into “Asteraceae” in Line 387.

Remark: Line 167-169: the sentence “For biotic stress, common sunflower cultivars TK0409 (susceptible) and JY207 (resistant), and root parasitic weed Orobanche cumana were applied” is not clear. Do you mean: “For biotic stress, root parasitic weed Orobanche cumana were applied to common sunflower cultivars TK0409 (susceptible) and JY207 (resistant)”?

Answer: Thanks for your kind suggestion. We have changed this sentence into “For biotic stress, root parasitic weed Orobanche cumana were applied to common sunflower cultivars TK0409 (susceptible) and JY207 (resistant)” in Line 169-171.

Remark: Line 194-203: please also indicate if house-keeping genes were used as reference for the qPCR, and if yes, what genes are used?

Answer: Thanks for your kind comment. ACT1, a house-keeping gene was used as reference for the qPCR. In the revised manuscript, we have added “and the ACT1 was selected as the reference gene.” after “The 2−ΔΔCt method with three replications was performed for analysis” in Line 202-203. We also have revised accordingly in the Supplementary Table S1.

Remark: Line 332: please change “23” into “Twenty-three”

Answer: Thanks for your kind suggestion. We have changed the number “23” into “Twenty-three” in Line 334.

Remark: Line 370: please delete “fortunately”

Answer: Thanks for your kind suggestion. We deleted “fortunately” in Line 378 between “and” and “we”.

Remark: Line 375: please delete “results” at the end of the line

Answer: Thanks for your kind suggestion. We deleted “results” at the end of Line 383.

Remark: Conclusion section: please also indicate here what HaWRKY genes (and/or how many) are the most important for each of the biotic or abiotic stresses.

Answer: Thank you for your kind comment. In the revised manuscript, we have tried our best to make the conclusion more specific. The sentence “The expressions of HaWRKY genes suggested that these genes could be involved in the regulation of sunflower growth and development, as well as various abiotic and biotic stresses” has been changed into “The expressions of HaWRKY genes suggested that most of the 23 selected HaWRKY genes were found to play an important role in regulating PEG and salt stress and 9 genes including HaWRKY 7/15/44/45/68/71/72/76/85 from 90 HaWRKY genes have potential roles against O. cumana infection” in Line 495-498.

Remark: Fig. 4 legend: “On the right side, Different motifs…” please change “Different” into “different”.

Answer: Thanks for your kind comment. The legend of Fig. 4 “On the right side, Different motifs…” has been changed into “On the right side, different motifs…”.

Comments of Reviewer #2: In the manuscript “Genome-wide characterization of WRKY gene family in Helianthus annuus L. and their expression profiles under biotic and abiotic stresses”, Li et al. identified WRKY family genes in sunflower and renamed them according to their locations on 17 chromosomes. They classified them into four main groups including a species-specific WKKY group using phylogenetic analyses, and they provided deep insight to the evolution of HaWRKY genes in sunflower by synteny analyses. Through transcriptomic and qRT-PCR analyses, they displayed distinct expression patterns of HaWRKY genes in different tissues, and under various abiotic and biotic stresses, which provide a foundation for further functional analyses of these genes.

This manuscript provides compelling evidence that HaWRKY family genes are associated with the development of the sunflower, as well as various abiotic and biotic stresses responsive, and provides a foundation for further functional analyses of HaWRKY genes. By using a non-standard genetic model like sunflower, the authors also help diversify studies in plant development. Please find my comments on the manuscript below.

Remark 1: Abstract is generally well-written but lacks some important data of the results.

Answer: Thanks for your kind suggestion. We have added the important results in the Abstract Section such as “Sunflower (Helianthus annuus L.) is one of the important vegetable oil supplies in the world.” In Line 23-24, and “The dN/dS ratio of these duplicated gene pairs were also calculated to understand the evolutionary constraints.” In Line 35-36.We also have revised “The tandem and segmental duplication possibly contributed to the diversity and expansion of HaWRKY gene families” into “The gene duplication analysis showed that five pairs of HaWRKY genes (HaWRKY8/9, HaWRKY53/54, HaWRKY65/66, HaWRKY66/67 and HaWRKY71/72) are tandem duplicated and four HaWRKY gene pairs (HaWRKY15/82, HaWRKY25/65, HaWRKY28/55 and HaWRKY50/53) are also identified as segmental duplication events, indicating that these duplication genes were contribute to the diversity and expansion of HaWRKY gene families in Line 30-35. 

Remark 2: Figure 7 and 8: Student’s t-test is fine here, but I suggest using Tukey’s HSD test to test all comparisons. This will help to show if 10% and 20% PEG, as well as 15mM and 30mM NaCl lines are different from each other, rather than just comparing to CK.

Answer: Thanks for your kind comment. In the revised manuscript, we have reanalyzed the significant expression levels of different HaWRKY genes under different treatments carefully by Tukey’s HSD test in Fig. 7 and Fig. 8. It is important to test all comparisons, especially for different concentrations of PEG and NaCl treatments. We also have changed “Student’s t-test” into “Tukey’s HSD test” in Materials and Methods section in Line 204.

Remark 3: Line 53 “Functional analyses show that WRKY…” should be “Functional analyses showed that WRKY g…”.

Answer: Thanks for your kind suggestion. The sentence “Functional analyses show that WRKY…” has been changed into “Functional analyses showed that WRKY g…”in Line 60.

Remark 4: In Materials and Methods part, some paragraph could be more concise such as plant materials, growth conditions and treatments section.

Answer: Thanks for your kind comment. We have revised “The seeds were germinated and grown in peat moss. At the five-leaf stage, seedlings with uniform size were selected for hydroponics in a half-strength Hoagland nutrient solution (pH=6.0). ” into “The seeds were germinated and grown in peat moss according to our previous study [32].” in Line 162 and cited the related literature. We also changed “The treatment concentrations were selected based on our preliminary experiments, in which 0, 100, 200, 300, 400 and 600 mM of NaCl were used. Little damage, significant damage and plant death were observed when the concentration of NaCl reached 100 mM, 300 mM, and over 300 mM, respectively. The concentrations of simulated drought stress were selected similarly.” into “The treatment concentrations were selected based on our preliminary experiments [32].” in Line 165-166.

Remark 5: The English writing should be substantially improved. Many sentences are illogical and cannot be understandable.

Answer: Thanks for your kind suggestion. We have tried our best to improve the manuscript. We revised “Data were transformed with a log2 (FPKM+1) transformation” into “Data were transformed by log2 (FPKM+1)” in Line 186. 

We deleted “in this WKKY group” between “distributed genes” and “into group I, II and III” in Line 224.

We added “ten” between “all” and “tissues” in Line 320, which could make the expression much more specific. 

We have substantially improved the manuscript, especially the Results section in Line 339-354, which will make the manuscript much more clear and understandable.

We also have changed “We identified 90 WRKY genes from Helianthus annuus L. and they were classified into four main groups including a species-specific WKKY group.” into “In this study, we identified 90 WRKY genes from Helianthus annuus L. and characterized their structure, duplication, chromosomal distribution, phylogenetic tree, followed by tissue-differential gene expression and differential expression in response to biotic and abiotic stress.” in Line 489-492.

We have revised “This study provides a foundation for further functional analyses of HaWRKY genes” into “Taken together, this study provides a good basis for further investigation of the biological functions and evolution of HaWRKY genes” in Line 498-499 in Conclusion section.

For detail, please refer to the revised manuscript in red front.

Remark 6: Overall: Add test statistics and p-values where significance is mentioned in text.

Answer: Thanks for your kind comment. We have added the test statistics and p-values where significance is mentioned in text. We changed “significantly up-regulated” into “significantly (P < 0.05) up-regulated by 46%/140% and 70%/51% compared with control, respectively.” In Line 339-340. We revised “no any changes of HaWRKY55/57 were observed in response to abiotic stresses” into “But no any significant (Tukey, P < 0.05) changes of HaWRKY55/57 were observed in response to abiotic stresses in all comparisons” in Line 340-341. We also changed “among them HaWRKY22/29/74 were significantly down-regulated only at high concentration of PEG. HaWRKY57/59/77/81/82/89 were significantly induced after treatment with PEG and HaWRKY30/35/38/48/55 were not sensitive to PEG in the roots.” into “For example, HaWRKY13/14/16/57/77 were significantly (P < 0.05) down-regulated under 10% and 20% drought stress, while HaWRKY29/74 were significantly (P < 0.05) down-regulated by 55% and 70% only at high concentration of PEG, as compared with control. The expression level of HaWRKY52 and HaWRKY89 were up-regulated by PEG, and 10% PEG significantly (P < 0.05) increased by 152% and 179% compared with control” in Line 343-349.

Remark 7: Overall: Italicize gene names.

Answer: Thanks for your kind suggestion. We have carefully checked all name of genes and italicized gene names in Line 336, 462, 573, 580, 583, 593, 619, 676, 678, and Line 683. We also have revised gene names accordingly in the Table 1 captions and the Supplementary Table S1 according to your suggestion.

---

## [Decision Letter · Decision Letter 1]

26 Oct 2020

Genome-wide characterization of WRKY gene family in Helianthus annuus L. and their expression profiles under biotic and abiotic stresses

PONE-D-20-29683R1

Dear Dr. Ling XU,

We’re pleased to inform you that your manuscript has been judged scientifically suitable for publication and will be formally accepted for publication once it meets all outstanding technical requirements.

Kind regards,

Basharat Ali, Ph.D

Academic Editor

PLOS ONE

Additional Editor Comments (optional):

Reviewers' comments:

Reviewer's Responses to Questions

**Comments to the Author**

1. If the authors have adequately addressed your comments raised in a previous round of review and you feel that this manuscript is now acceptable for publication, you may indicate that here to bypass the “Comments to the Author” section, enter your conflict of interest statement in the “Confidential to Editor” section, and submit your "Accept" recommendation.

Reviewer #1: All comments have been addressed

Reviewer #2: All comments have been addressed

2. Is the manuscript technically sound, and do the data support the conclusions?

Reviewer #1: Yes

Reviewer #2: Yes

3. Has the statistical analysis been performed appropriately and rigorously? 

Reviewer #1: Yes

Reviewer #2: Yes

4. Have the authors made all data underlying the findings in their manuscript fully available?

Reviewer #1: Yes

Reviewer #2: Yes

5. Is the manuscript presented in an intelligible fashion and written in standard English?

Reviewer #1: Yes

Reviewer #2: Yes

6. Review Comments to the Author

Reviewer #1: The authors have addressed all my review comments. I would recommend that the current version is ready for publication.

Reviewer #2: (No Response)

7. PLOS authors have the option to publish the peer review history of their article (what does this mean?). If published, this will include your full peer review and any attached files.

Reviewer #1: **Yes: **Hui Liu

Reviewer #2: No

---

## [Editor Report · Acceptance letter]

17 Nov 2020

PONE-D-20-29683R1 

Genome-wide characterization of *WRKY* gene family in *Helianthus annuus* L. and their expression profiles under biotic and abiotic stresses 

Dear Dr. Xu:

I'm pleased to inform you that your manuscript has been deemed suitable for publication in PLOS ONE. Congratulations! Your manuscript is now with our production department. 

Kind regards, 

on behalf of

Dr. Basharat Ali 

Academic Editor

PLOS ONE